# Filbertone Ameliorates Adiposity in Mice Fed a High-Fat Diet via Activation of cAMP Signaling

**DOI:** 10.3390/nu11081749

**Published:** 2019-07-30

**Authors:** Youna Moon, Tao Tong, Wesuk Kang, Taesun Park

**Affiliations:** Department of Food and Nutrition, Brain Korea 21 PLUS Project, Yonsei University, 50 Yonsei-ro, Seodaemun-gu, Seoul 03722, Korea

**Keywords:** filbertone, cAMP, adipogenesis, thermogenesis

## Abstract

The aim of this research was to estimate the preventive effects of filbertone, the main flavor compound in hazelnuts, on lipid accumulation in the adipose tissue of mice fed a high-fat diet (HFD) and to reveal the underlying molecular mechanisms. Male C57BL/6N mice were fed chow, a HFD, or a 0.025% filbertone-supplemented HFD for 14 weeks. We found that filbertone supplementation resulted in significant reductions in body weight gain and lipid accumulation in adipose tissue, with parallel improvements in plasma lipid levels (triglycerides, total cholesterol, and free fatty acids) and proinflammatory cytokines (interleukin 6 (IL-6) and tumor necrosis factor alpha (TNF-α)). Molecular analysis revealed that filbertone treatment led to reprogramming of metabolic signatures in the cyclic adenosine monophosphate (cAMP) pathway. Filbertone supplementation significantly increased the cAMP level and increased downstream protein kinase A catalytic subunit (PKA) signaling in mouse adipose tissue. The mRNA level of adipogenesis-related genes was downregulated in the adipose tissue of filbertone-fed mice compared to control mice fed the HFD alone. Furthermore, filbertone treatment elevated the expression of thermogenic genes in mouse adipose tissue. Filbertone reduced intracellular lipid accumulation and increased the oxygen consumption rate in 3T3-L1 cells and these filbertone-induced changes were abrogated by the adenylate cyclases (ADCY) inhibitor. Taken together, our results suggest that the beneficial effects of filbertone on lipid accumulation may be associated with the activation of cAMP signaling.

## 1. Introduction

Obesity is a chronic condition of energy imbalance whereby a long-term overload of energy intake over expenditure results in the stocking of that excess energy in white adipose tissue (WAT) [1]. Excess WAT is well established to be predisposed to the development of dyslipidemia, insulin resistance, diabetes, and accelerated macrovascular disease. As obesity progresses, the storage capacity of the adipose tissue becomes saturated, following which lipids accumulate at ectopic sites (non-adipose tissues) [2]. The existence of fat at ectopic sites, either alone or in relation to elevated visceral adiposity, is also an independent determinant of the development of a large number of metabolic diseases [3,4]. Despite the high prevalence of obesity and the enormous efforts in both academia and industry to develop anti-obesity medication, adequate and secure pharmacological options for the treatment and prevention of obesity remain elusive.

Cyclic adenosine monophosphate (cAMP), as important messenger, is pivotal in the control of energy homeostasis [5,6,7,8]. For example, signaling by β-adrenergic receptors elicits increased cAMP levels and subsequent lipolysis in WAT and brown adipose tissue (BAT); mice lacking all three β-adrenergic receptors (β1-, β2-, and β3-AR) have a decreased metabolic rate and are mildly obese when fed on chow and massively obese when fed on a high-fat diet (HFD) [9]. Specific deletion of Gn_S_α, the G protein required for receptor-stimulated cAMP production, leads to insulin resistance and obesity in mice [10]. In 3T3-L1 adipocytes, the rise in intracellular cAMP concentrations by treatment with an adenylyl cyclase activator, forskolin, downregulates the mRNA levels of major adipogenic transcription factors [11] and inhibits intracellular lipid accumulation, as checked by the morphological change. These studies have highlighted that enhancement of cAMP signaling may be helpful for the treatment or prevention of obesity and its related metabolic syndrome.

We recently undertook a phenotype-based screen in adipocytes for phytochemicals that could ameliorate intracellular lipid accumulation. In this screen, we found filbertone (C_8_H_14_O), a major flavor compound in the fruits of hazel trees (*Corylus avellana* and *Corylus maxima*) [12], to be a good candidate molecule warranting further investigation. Filbertone has been acknowledged as a commonly accepted safe chemical by the Flavor and Extract Manufacturers Association (FEMA) [13]. Although this compound is widely used as a food flavoring in nuts, coffee, cocoa, meat, mints, and tropical, citrus, or fruity foods [12], research on the physiological roles of filbertone is limited. In the present study, we aimed to test whether filbertone protects against HFD-induced adiposity in mice, and, if so, to clarify whether filbertone exerts its anti-obesity effect via activation of the cAMP pathway.

## 2. Materials and Methods

### 2.1. Reagents

Filbertone (#W376108), dimethyl sulfoxide (DMSO, #D2650) and Oil Red O solution (ORO, #O0625) were acquired from Sigma-Aldrich (St. Louis, MO, USA). The antibodies against, protein kinase A catalytic subunit (PKA Cα, cat. #4782, 1:1000), 5′-adenosine monophosphate-activated protein kinase (AMPK, cat. #2532, 1:1000), phospho-AMPK (cat. #2531; 1:1000), cAMP-responsive element-binding protein (CREB, cat. #9197, 1:1000), phospho-CREB (cat. #9198; 1:1000), tuberous sclerosis complex 2 (TSC2, cat. #4308, 1:1000), phospho-TSC2 (cat. #5584; 1:1000), mechanistic target of rapamycin (mTOR, cat. #2972; 1:1000), phospho-mTOR (cat. #2974; 1:1000), ribosomal protein S6 kinase (S6K, cat. #2708, 1:1000), phospho-S6K (cat. #9204; 1:1000), peroxisome proliferator-activated receptor gamma coactivator 1 alpha (PGC1α, cat. #2178, 1:1000), and glyceraldehyde-3-phosphate dehydrogenase (GAPDH) (cat. #2118; 1:5000) were obtained from Cell Signaling Technology (Danvers, MA, USA). The enzyme horseradish peroxidase (HRP)-labeled anti-rabbit IgG antibody (1:5000; Santa Cruz cat. # sc-2004; secondary antibody) was used for western blotting.

### 2.2. Animal Experiments

Five-week-old male C57BL/6N mice were obtained from Orient Bio Inc. (Seongnam, Gyeonggi-do, Korea) and housed in a temperature- (21 °C) and humidity- (55%) controlled environment with a 12 h–12 h light–dark cycle (lights on at 08.00 h). The mice were given free access to tap water and a commercially available diet (chow, Orient Bio Inc., Seongnam, Korea) ad libitum for one week prior to their distribution in three groups (*n* = 8 per group), namely, the chow, HFD (40% fat), and filbertone-supplemented diet groups. The filbertone-supplemented diet was identical to the HFD but also involved 0.025% filbertone. The mice were subjected to the experimental diet for 14 weeks, during which time all animals were permitted ad libitum access to the diet and water. Food intake was measured daily and body weight was measured weekly. Mice were anesthetized with avertin after a 6 h fast at the end of the experiment. Blood samples were obtained from the abdominal aorta into an ethylenediaminetetraacetic acid (EDTA)-coated tube and centrifuged at 4000× *g* for 15 min. The visceral adipose tissues (epididymal, mesenteric, perirenal, and retroperitoneal fat) were removed, measured, instantly snap-frozen in liquid nitrogen, and kept at −80 °C until further analysis. All procedures were reviewed by the Institutional Animal Care and Use Committee (IACUC) of the Yonsei Laboratory Animal Research Center (permit no. IACUC-A-201706-587-02).

### 2.3. Oral Glucose Tolerance Test

The oral glucose tolerance test (OGTT) was measured on 6 h fasted mice two weeks prior to the termination of experiment. After measuring basal blood glucose concentrations in tail vein blood with a One Touch Ultra Blood Glucose Meter (LifeScan Inc., Milpitas, CA, USA), the mice were given a glucose load of 2 g/kg body weight through oral gavage and blood glucose concentrations were checked again 15, 30, 60, 90, and 120 min after glucose administration.

### 2.4. Histopathological Examinations

Formalin-fixed epididymal adipose tissues were embedded in paraffin and sections were cut at 5 µm prior to staining with hematoxylin and eosin (H&E). All histological processes were handled at the Korea CFC pathology laboratory (Yongin-si, Gyeonggi-do, Korea). The stained tissue pieces were viewed under bright field microscopy at ×200 magnification, and images were acquired using a DP-70 camera and DP controller software (Olympus, Tokyo, Japan). The cross-sectional diameters of adipocytes were measured with ImageJ software (National Institute of Health, Bethesda, MD, USA).

### 2.5. Biochemical Analysis

Plasma was collected by centrifugation (15 min, 4000× *g*, 4 °C) in an EDTA-coated tube. All plasma samples were snap frozen and stored at −80 °C. The plasma levels of glucose, free fatty acids, triglycerides, and total cholesterol were determined using commercially available kits (Bio-Clinical System, Gyeonggi-do, Korea). The plasma concentrations of insulin, tumor necrosis factor-alpha (TNF-α), monocyte chemoattractant protein 1 (MCP1), interleukin 6 (IL-6), adiponectin, and leptin were estimated using a commercially available enzyme-linked immunosorbent assay (ELISA) kit (Millipore, Burlington, MA, USA). The absorbance was measured using a microplate spectrophotometer (Tecan, Männedorf, Switzerland).

### 2.6. cAMP Assay

cAMP levels in epididymal adipose tissues or 3T3-L1 adipocytes were evaluated using a colorimetric cAMP ELISA kit (Enzo Life Sciences, Farmingdale, NY, USA), in accordance with the manufacturer’s suggested instructions. Briefly, filbertone or vehicle-treated 3T3-L1 adipocytes were homogenized in 0.1 N HCl for 10 min, centrifuged at 1000× *g* for 5 min, and 100 µL of supernatant (cell lysate) was then transferred for ELISA assay. The cAMP levels were determined via estimation of the optical density (OD) at 410 nm with a Tecan plate reader (Männedorf, Switzerland). Results were standardized to the protein content present in the cell lysate as performed by the bicinchoninic acid (BCA) assay.

### 2.7. Western Blot Assay

To obtain total protein, the epididymal tissues of mice was lysed in a buffer containing 100 mM Tris-HCl (pH 7.4), 100 mM orthovanadate, 50 mM NaF, 50 mM sodium pyrophosphate, 5 mM EDTA, 1% Triton X-100, 1 mM phenylmethanesulfonyl fluoride, 2 µg/mL aprotinin, 1 µg/mL pepstatin A, and 1 µg/mL leupeptin. The tissues were vortexed for 15 min on ice and centrifuged for 15 min at 1200× *g*. The protein concentration of each supernatant was quantified using the Bradford assay reagent (Bio-Rad, Hercules, CA, USA) following the manufacturer’s protocols. Lysate samples consisting of 35 µg of protein were added to a SDS-PAGE loading buffer and then loaded onto a 10% sodium dodecyl sulfate–polyacrylamide gel. Antibodies were as follows: GAPDH, PKA Cα, AMPK, phospho-AMPK (Thr172), TSC2, phospho-TSC2 (Ser1387), S6K, phospho-S6K (Thr389), mTOR, phospho-mTOR (Ser 2448), PGC1α, CREB, and phospho-CREB (Ser133); all the antibodies were purchased from Cell Signaling Technology (Danvers, MA, USA). Finally, the enhanced chemiluminescence (ECL) (GE Healthcare, Amersham, UK) was used to image the protein bands. Images were acquired with a LuminoGraph system (WSE-6100, Atto, Tokyo, Japan).

### 2.8. Quantitative Real-Time PCR

Total RNA was lysed from epididymal fat with TRIzol (Invitrogen, Carlsbad, CA, USA). cDNA synthesis was conducted with total RNA (1 µg), dithiothreitol (0.1 M, Invitrogen), RNase inhibitor (40 U/µL, Invitrogen), reverse transcriptase (200 U/µL, Invitrogen), dNTPs (2.5 mM, Invitrogen), and 1 × Reverse transcriptase (RT) buffer (Invitrogen) in a 40 µL volume at 37 °C. Quantitative real-time PCR was used with the CFX Connect Real-Time PCR Detection System (Bio-Rad) using SYBR green master mix (Bio-Rad). Primer sequences are listed in Table 1.

### 2.9. Cell Culture and Treatment

3T3-L1 preadipocytes were obtained from American Type Culture Collection (ATCC Manassas, VA, USA) and cultured in a humidified atmosphere with 5% CO_2_ at 37 °C. The cells were differentiated into adipocytes as previously reported [14]. Briefly, 3T3-L1 preadipocytes were cultured in Dulbecco’s modified Eagle’s medium (DMEM) containing 10% bovine calf serum (Gibco life technologies, Carlsbad, CA, USA) and 1% antibiotic–antimycotic solution (Gibco life technologies). For the differentiation assay, preadipocytes were seeded in six-well plates and cultured until two days post-confluence. After the 3T3-L1 cells became confluent, the medium was replaced with DMEM consisting of 10% of fetal calf serum (FBS; Gibco-BRL), 1% of antibiotic–antimycotic solution, 10 μg/mL bovine insulin, 1 μM dexamethasone (Sigma-Aldrich, St. Louis, MO, USA), and 0.5 mM isobutylmethylxanthine (Sigma-Aldrich) with or without filbertone (Sigma-Aldrich, St. Louis, MO, USA) and SQ22,536 (an adenylate cyclases (ADCY) inhibitor, Sigma-Aldrich, St. Louis, MO, USA). The cells were then maintained in DMEM containing 10% of FBS with 10 μg/mL insulin for another two days, followed by culturing with DMEM containing 10% of FBS for an additional four days with or without filbertone and SQ22,536.

### 2.10. MTT Assay

To evaluate the toxicity of filbertone, 3T3-L1 preadipocytes were plated into 96-well microtiter plates at a density of 1 × 10^4^ cells/well. After 24 h, the culture medium was replaced by 200 µL serial dilutions (0–150 µM) of filbertone and the cells were incubated for another 24 h. The culture solutions were then removed and replaced by 90 µL of culture medium and 10 µL of filtered MTT solution (10 mg/mL) in phosphate-buffered saline (PBS) to reach a final concentration of 1 mg MTT/mL. After 3 h, the unresponded dye was removed, and then the insoluble formazan crystals were dissolved in DMSO (50 µL/well) and measured spectrophotometrically in a microplate reader (Tecan, Männedorf, Switzerland) at 570 nm.

### 2.11. Oil Red O Staining

Differentiated 3T3-L1 cells were washed with PBS, fixed with 4% paraformaldehyde solution (Cellnest, Minato, Japan) for 20 min and allowed to dry. Lipid droplets in mature adipocytes were then stained with Oil Red O staining solution (0.5% Oil red O in isopropanol) for 20 min and washed with distilled water. Microscope images were obtained to assess the stained oil droplets. To quantify the staining intensity, isopropanol was added into each well to dissolve the staining and the dissolved solution was transferred to another 96-well plate. The OD was measured at a wavelength of 500 nm using a microplate reader (Molecular Devices, San Jose, CA, USA).

### 2.12. Cellular Metabolic Rates

The mitochondrial oxygen consumption rate (OCR) in 3T3-L1 cells was estimated with a Seahorse XF-24 analyzer (Seahorse Bioscience, North Billerica, MA, USA). Briefly, 3T3-L1 cells were seeded into XF-24 microplates and induced to differentiate with 5% CO_2_ at 37 °C. The cells were maintained in a non-CO_2_ incubator for 1 h before assay. The OCR was standardized to protein content. Each experiment was conducted with five replicates.

### 2.13. Statistical Analyses

Statistical analysis was performed using SPSS 24.0 software (SPSS Inc., Chicago, IL, USA). Statistical analysis of the data was conducted using unpaired Student’s *t*-test. Data are presented as means ± SEM. Data describing body weight gains, visceral fat-pad weights, plasma characteristics, and OGTT are presented as means ± SEM of eight mice per group. The RT-PCR and western blot results are means from an *n*  =  8 ± SEM of three independent experiments (*n* = 2 or 3 per experiment) for each group. A *p*-value ≤ 0.05 was considered statistically significant and significance was set at * *p* < 0.05, ** *p* < 0.01, and *** *p* < 0.001.

## 3. Results

### 3.1. Filbertone Reduced Both HFD-Induced Weight Gain and Adiposity

Filbertone treatment for 14 weeks affected body weight compared to the HFD-only controls. Mice fed the filbertone for 14 weeks showed significant reductions in final body weight and body weight gain as compared to control mice fed HFD alone, without changes in food intake (Figure 1a–c). The total visceral (epididymal, perirenal, mesenteric, and retroperitoneal) fat pad weights were significantly lower in the filbertone-fed mice than in the HFD control mice (Figure 1d). We next examined adipocyte size to determine whether filbertone inhibits lipid storage in adipose tissues. Filbertone treatment resulted in significantly reduced adipocyte size compared to the HFD-only group (Figure 1e). By week 14 of diet treatment, filbertone markedly inhibited the HFD-induced expansion of both body weight and visceral fat mass.

### 3.2. Filbertone Improved Obesity-Related Biochemical Parameters

To estimate whether filbertone could enhance glucose metabolism in vivo, we first treated mice with vehicle or filbertone (30 mg/kg) and conducted an oral glucose tolerance test. Following the exogenous glucose challenge, glucose levels were lower in filbertone-treated mice than in control mice fed the HFD alone, and the area-under-the-curve (AUC) value from 0 to 120 min was also lower, indicative of improved glucose tolerance (Figure 2a). Fasting glucose and insulin concentrations were significantly lower in filbertone-treated mice than in HFD control mice (Figure 2b,c). The plasma levels of inflammatory cytokines in filbertone- and HFD-treated mice are summarized in Figure 2d–f. Mean values for IL-6 (Figure 2d), TNF-α (Figure 2e), and MCP1 (Figure 2f) were lower in filbertone-treated mice than in those treated with a HDF alone. We compared plasma adiponectin and leptin levels between filbertone-treated mice and those treated with a HFD alone. The plasma adiponectin level was 130% higher in filbertone-treated mice than in those in the HFD-only group, whereas that of leptin was 25% lower (Figure 2g,h). Compared to the HFD-only group, filbertone-supplemented mice demonstrated a decrease in the plasma level of triglycerides, free fatty acids, and cholesterol (Figure 2i–k).

### 3.3. Filbertone Activated the cAMP-PKA Signaling Pathway in Adipose Tissue

The cAMP concentration in the adipose tissue was significantly higher in filbertone-treated mice than in HFD control mice (Figure 3a). Filbertone elevated the protein expression of PKA Cα and p-AMPK (Figure 3b). Filbertone stimulated the phosphorylation of TSC2, mTOR, and S6K (Figure 3c) in mouse epididymal adipose tissue, demonstrating that the adipogenesis signaling pathway was activated. According to the RT-PCR analysis of the adipose tissue, genes, such as CCAAT/enhancer binding-protein α (*C/EBPα)*, adipocyte fatty acid binding protein (*aP2)*, and peroxisome proliferator-activated receptor γ2 (*Pparγ2)* involved in adipogenesis were markedly downregulated by filbertone treatment (Figure 3d). Notably, p-CREB and PGC1α, two transcriptional regulators that specify the oxidative metabolic gene expression program in adipose tissue, were also upregulated by filbertone inclusion in the HFD (Figure 4a). Moreover, we detected increased *Pgc1α*, PR domain containing 16 (*Prdm16)*, cell death activator CIDE-A (*Cidea)*, and uncoupling protein 1 (*UCP1)* mRNA expression levels in the epididymal tissue from filbertone-treated mice compared to control mice fed the HFD alone (Figure 4b). Taken together, filbertone induced activation of the cAMP–PKA pathway is a likely mechanism contributing to anti-adipogenic and thermogenic effects of filbertone (Figure 5). 

### 3.4. Filbertone Inhibited 3T3-L1 Adipocyte Differentiation by Upregulating cAMP

The effect of filbertone on the viability of 3T3-L1 preadipocytes was estimated with redox activity in live cells by MTT assay. The 3T3-L1 cells treated with filbertone (25–150 µM) for 24 h exhibited a significant concentration-dependent effect compared to the untreated control cells. To examine the effects of filbertone on adipocyte differentiation, density-arrested 3T3-L1 preadipocytes were treated with different doses of filbertone and stimulated to differentiate with methylene diphenyl isocyanate (MDI). Filbertone dose-dependently inhibited adipocyte differentiation (Figure 6a–c). The increased expression of thermogenic genes such as *PGC1-α*, *PRDM16*, *Cidea*, and *UCP1* observed in filbertone-fed mice could be associated with activation of cAMP-mediated pathways. Indeed, we observed that filbertone treatment increased the cAMP level 1.8-fold in 3T3-L1 adipocytes (Figure 6d). We found that the positive effects exerted by filbertone on the reduction of lipid accumulation were lost in the presence of the adenylate cyclase inhibitor SQ22,536 (9-(tetrahydrofuryl)-adenine) (Figure 6e). Analysis of the oxygen consumption rate revealed a significant increase in the basal respiration rate of filbertone-treated preadipocytes compared to control cells (Figure 6f). 

## 4. Discussion

The current dose of filbertone (25 mg/(kg·d)) used in mice corresponds to approximately 2 mg/(kg·d) in humans when measured based on standardization to the body surface area, as recommended by Reagan-Shaw et al. [15]. In our preliminary study, 10, 25, or 50 mg/(kg·d) filbertone was orally administered to C57BL/6N mice once a day for 28 days. The minimal effective dose of filbertone that resulted in decreased body weight was found to be 25 mg/(kg·d), with no further reduction being seen with 50 mg/(kg·d). Oral administration dose of 25 mg filbertone/kg body weight equals 1 mg filbertone/mouse (40 g body weight). Assuming that mice consumed 1 mg filbertone in 4 g diet, the diet should contain 0.025% filbertone.

In the present study, we found that filbertone decreases adipogenic metabolism in visceral fat and 3T3-L1 cells likely through the activation of the cAMP pathway. Coincident with this decrease in visceral fat content, filbertone also elicited a decrease in the expression levels of adipogenic-related proteins and genes. cAMP pathways are recognized as being crucial for the inhibition of adipocyte development [16,17]. Binding of cAMP to the PKA regulatory subunit results in the release of the catalytic subunit, which can then phosphorylate its lipid metabolism-associated protein substrates, including AMPK, in all types of adipocytes [18]. Phosphorylated AMPK inhibits preadipocyte differentiation by downregulating *C/EBPα* and *PPARγ*, the main regulators of adipogenesis and lipid store in adipocytes [19,20].

The CREB protein acts as a transcription factor that binds to cAMP response elements (CREs) within the promoters of its target genes when phosphorylated at Ser133 by different receptor-activated protein kinases such as PKA [21]. In adipocytes, it has been confirmed that increase of CREB activity via the cAMP–PKA pathway stimulates the level of PGC1α; this activates the transcription of thermogenic genes, such as *PRDM16*, *Cidea*, and *UCP1* [22]. Here, coinciding with significant upregulation of p-CREB, increased *PGC1-α* and *UCP1* mRNA expression was observed in the adipose tissues of filbertone-treated mice compared to the HFD control group (Figure 3). Furthermore, the capacity of filbertone to induce oxygen consumption in adipocytes was strongly attenuated in the presence of a cAMP inhibitor (Figure 3a,b). These results indicate that filbertone may affect energy expenditure in a cAMP-dependent mechanism.

The cAMP pathway typically includes the activation of a G protein-coupled receptor (GPCR) at the plasma membrane and subsequent activation of adenylyl cyclase (AC) which changes ATP to cAMP [5]. Olfactory receptors (ORs) constitute the largest subfamily of GPCRs and were originally assumed to exist exclusively in the olfactory epithelium [23]. Nevertheless, recent research has shown that ORs are considerably more multifaceted and ORs are now considered as broad chemoreceptors that are present in diverse tissues, where they conduct various regulatory functions [24,25,26,27]. Moreover, it was newly revealed that in several nonchemosensory cells, such as adipocytes [28] and keratinocytes [29], activation of ectopically expressed ORs also results in ectopic OR pathway. These studies make ORs possible therapeutic targets in human disorder, in addition to their usefulness for the fragrance industry. Thus, we deduced that the defensive effect of filbertone against adiposity could be mediated by the activation of ORs expressed in adipose tissue. To identify the molecular targets of filbertone, we researched the molecular diversions occurring in adipose tissue in response to filbertone treatment using a mode-of-action by network identification (MNI), a validated method for the recognition of targets and related signaling of molecules [30,31]. The analysis of gene profiles with the highest MNI rankings and fold changes in differential expression led to the recognition of various ORs including Olfr 1034, 402, 508, and 110 as promising candidates. Identifying the specific OR accountable for the filbertone-induced elevation in cAMP levels in adipose tissue is a significant and challenging area for future work.

In the present study, filbertone treatment led to a significant improvement in glucose homeostasis in HFD-induced obese mice, as evidenced by the results of the glucose tolerance tests, as well as fasting blood insulin and glucose levels. The present results support that the marked attenuation of glucose homeostasis by filbertone inclusion in the HFD was possibly mediated by reduced visceral fat accumulation [32]. Insulin excitation of glucose uptake in fat and muscle tissues is one of the main activities of the hormone, which is essential for whole-body glucose homeostasis. There is increasing recognition that glucose homeostasis can be regulated through the modulation of GPCRs such as adrenoceptors in adipose tissue and skeletal muscle [33,34]. For example, β-adrenergic receptors in adipose tissue and skeletal muscle are primarily Gαs coupled and improve glucose tolerance through stimulation of AC and secretion of the cAMP. Although the mechanism is not clear, camp-induced PKA upregulated phosphorylated mammalian target of rapamycin complex 2 (mTORC2), which is needed for glucose uptake [35]. Further studies are needed to confirm whether the filbertone-associated improvement in glucose homeostasis is mediated by insulin-dependent signaling.

## 5. Conclusions

We found that filbertone, a naturally occurring flavoring agent, significantly improves visceral adiposity through activation of cAMP-mediated signaling cascades in WAT of mice fed a HFD. Although the effects of filbertone in humans have yet to be explained, the current study highlights the possible importance of filbertone for the prevention of obesity and its related metabolic complications.

## Figures and Tables

**Figure 1 nutrients-11-01749-f001:**
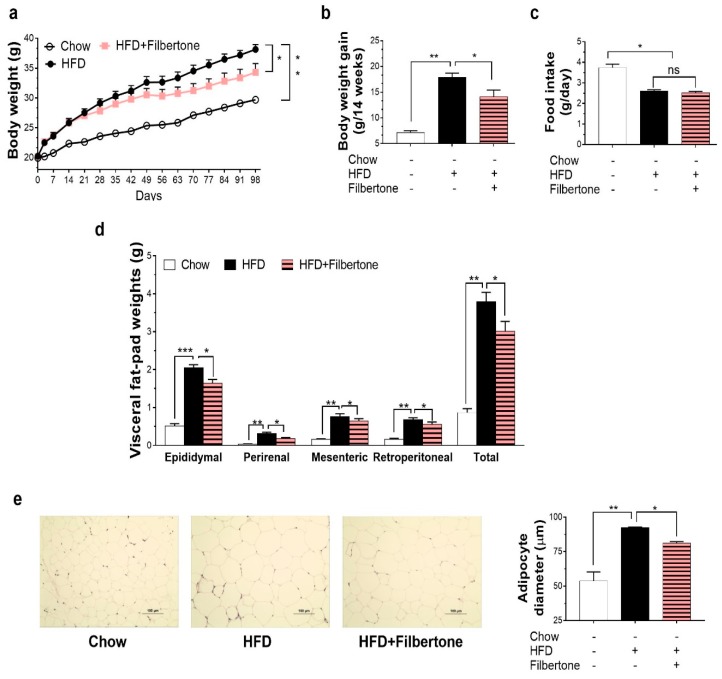
Effects of filbertone on obesity phenotypes in mice fed a high fat diet (HFD). Male C57BL/6N mice were fed chow, a HFD, or a HFD supplemented with filbertone for 14 weeks. Changes in (**a**) body weight, (**b**) body weight gain, and (**c**) food intake. (**d**) Visceral fat pad weights and representative images of visceral fat pads. Representative hematoxylin and eosin (H&E) staining of (**e**) epididymal fat. (**e**) The cross-sectional diameter of adipocytes was calculated using ImageJ; * *p* < 0.05, ** *p* < 0.01, and *** *p* < 0.001, between the groups. Values are means ± SEM.

**Figure 2 nutrients-11-01749-f002:**
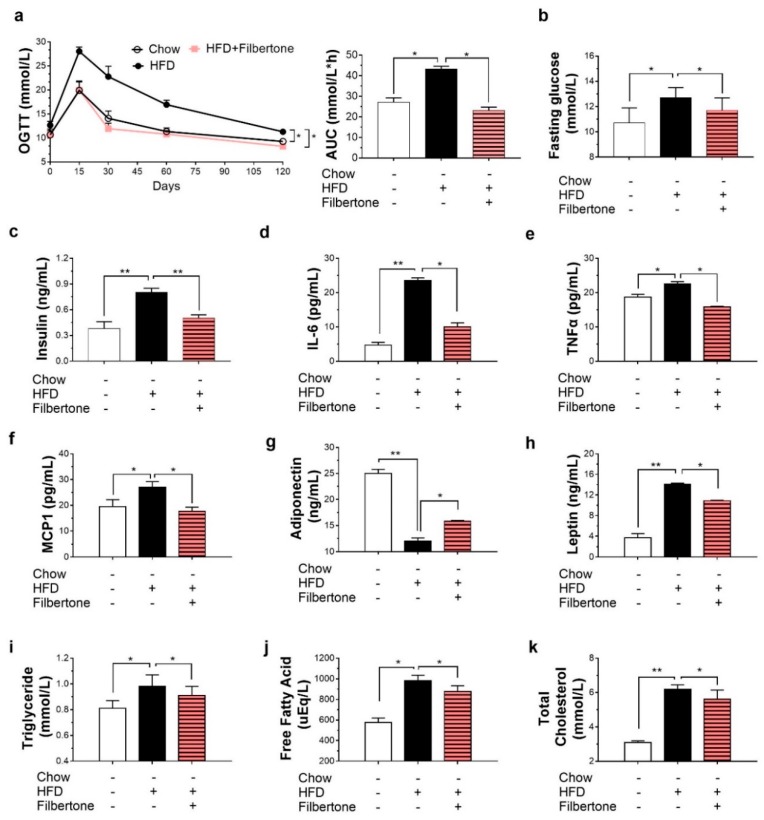
Mouse plasma biochemical parameters. (**a**) Oral glucose tolerance test (OGTT) and area under the curve (AUC); fasting plasma levels of (**b**) glucose and (**c**) insulin index; plasma levels of the proinflammatory cytokines (**d**) interleukin 6 (IL-6), (**e**) tumor necrosis factor alpha (TNF-α), (**f**) monocyte chemoattractant protein 1 (MCP1), (**g**) adiponectin, and (**h**) leptin. Plasma levels of (**i**) triglycerides, (**j**) free fatty acids (FFAs), and (**k**) total cholesterol * *p* < 0.05 and ** *p* < 0.01, between the groups. Values are means ± SEM.

**Figure 3 nutrients-11-01749-f003:**
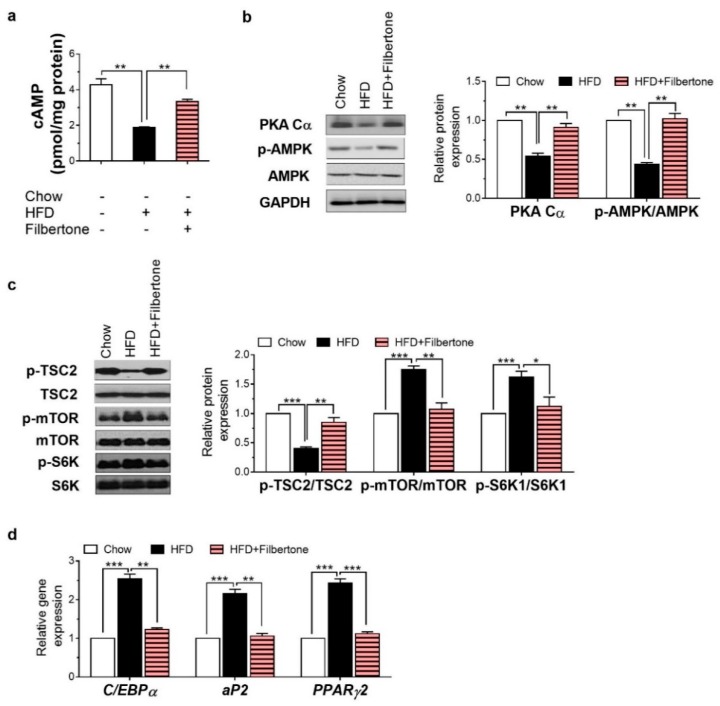
Filbertone regulates the expression of adipogenesis-related factors in mouse adipose tissue. (**a**) Cyclic adenosine monophosphate (cAMP) concentrations in mouse epididymal adipose tissue. (**b**) Protein levels of protein kinase A catalytic subunit (PKA Cα), phosphorylated adenosine monophosphate activated protein kinase (p-AMPK), and adenosine monophosphate activated protein kinase (AMPK) in mouse adipose tissue. (**c**) Protein levels of phosphorylated tuberous sclerosis complex 2 (p-TSC2), tuberous sclerosis complex 2 (TSC2), phosphorylated mechanistic target of rapamycin (p-mTOR), mechanistic target of rapamycin (mTOR), phosphorylated ribosomal protein S6 kinase 1 (p-S6K1), and ribosomal protein S6 kinase 1 (S6K1) in mouse adipose tissue. (**d**) Real time RT-PCR analysis of *C/EBPα*, *aP2*, and *PParγ2* in mouse adipose tissue. * *p* < 0.05, ** *p* < 0.01, and *** *p* < 0.001, between the groups. Values are means ± SEM.

**Figure 4 nutrients-11-01749-f004:**
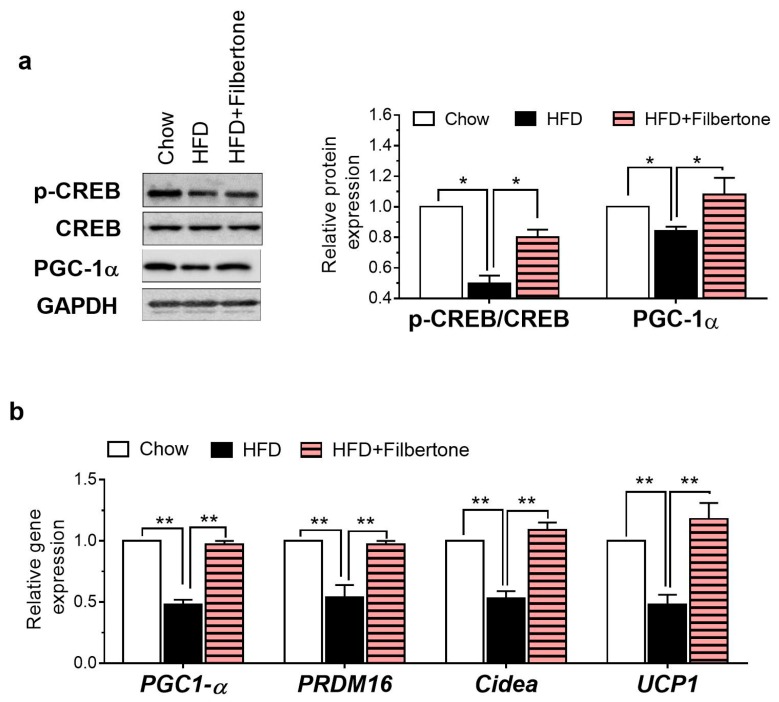
Filbertone regulates the expression of thermogenesis-related factors in adipose tissue. (**a**) Protein levels of phosphorylated cAMP-responsive element-binding protein (p-CREB), cAMP-responsive element-binding protein (CREB), and PGC1α in the adipose tissue. (**b**) Real time RT-PCR analysis of *PGC1-α*, *PRDM16*, *Cidea*, and *UCP1* in the adipose tissue. * *p* < 0.05 and ** *p* < 0.01, between the groups. Values are means ± SEM.

**Figure 5 nutrients-11-01749-f005:**
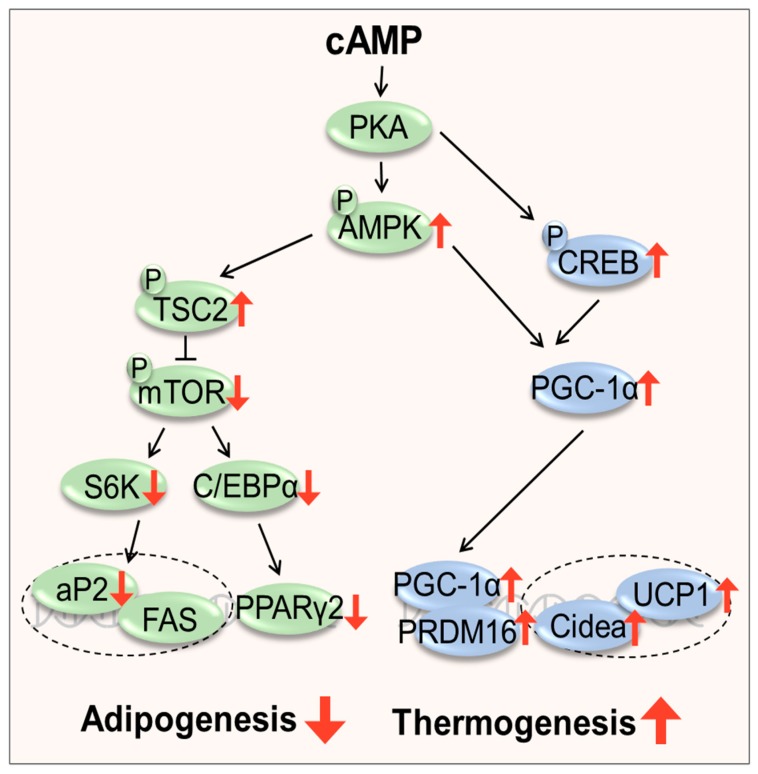
The proposed mechanism behind the protective effects of filbertone against adiposity in mice. High cytosolic levels of cAMP induced by filbertone activate PKA and thus cause phosphorylation of AMPK and CREB. Filbertone-induced AMPK activation inhibits adipocyte differentiation and blocks the expression of late adipogenic markers such as *fatty acid synthase (FAS)* and transcription factors, *PPAR-γ2* and *C/EBPα*, thereby inhibiting adipogenesis. Filbertone-induced CREB activation promotes the transcription of genes involved in thermogenesis, e.g., *PGC-1α* and *UCP1*, resulting in increased thermogenesis in adipocytes.

**Figure 6 nutrients-11-01749-f006:**
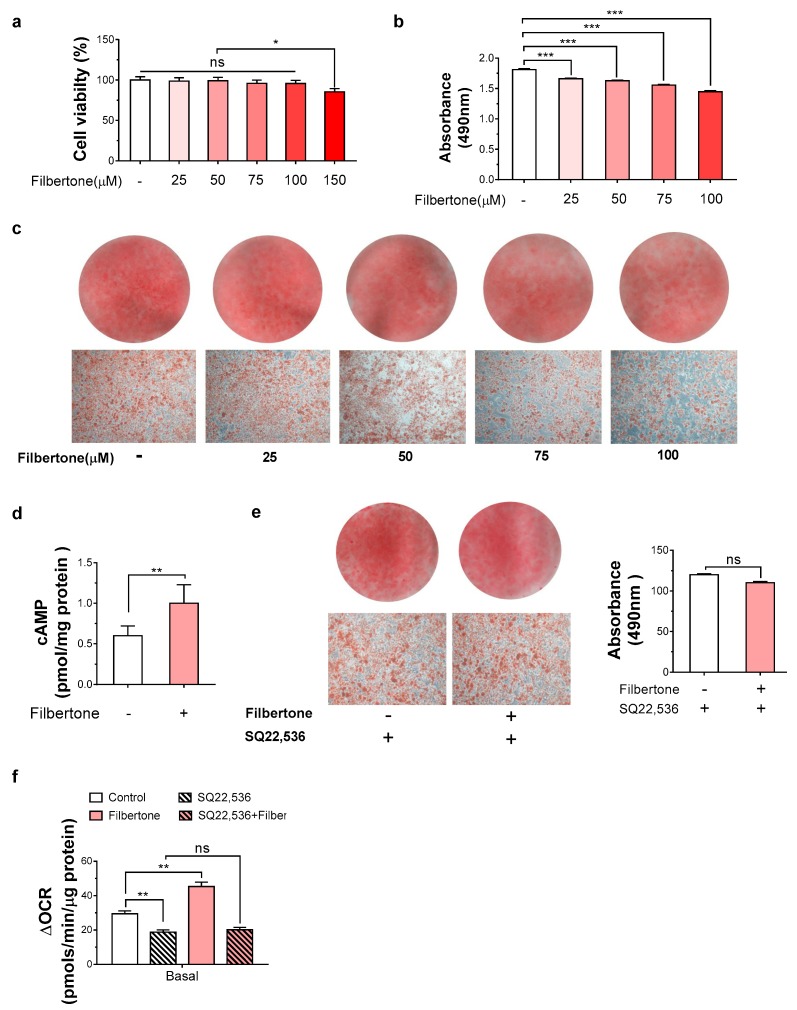
Filbertone reduces triglyceride accumulation in 3T3-L1 cells via the cAMP pathway. (**a**) The percentage of viable cells was performed by an MTT assay. (**b**,**c**) Oil Red O (ORO) staining of 3T3-L1 cells after stimulation with filbertone at the indicated concentration. (**d**) The cAMP concentration in 3T3-L1 cells treated with or without 50 µM filbertone. (**e**) ORO staining of 3T3-L1 cells treated with filbertone (50 µM) in the presence or absence of the adenylate cyclase inhibitor, SQ22,536 (100 µM). Spectrophotometric data for ORO-stained adipocytes are means from three independent experiments for each group, and representative photomicrographs (×200) are shown. (f) The oxygen consumption rate (OCR) was determined in differentiated 3T3-L1 cells under basal conditions. * *p* < 0.05, ** *p* < 0.01, and *** *p* < 0.001, between the groups. Values are means ± SEM.

**Table 1 nutrients-11-01749-t001:** Primer sequences.

Type	Gene Description	Sequences (5′→3)
Mouse	CCAAT/enhancer binding-protein α (*C/EBPα*)	F: TCAGCTTACAACAGGCCAGG
		R: ACACAAGGCTAATGGTCCCC
	Adipocyte fatty acid binding protein (*aP2*)	F: CATGCGACAAAGGCAGAAAT
		R: GTTACAAGGCAAGGAAGGGC
	Peroxisome proliferator-activated receptor γ2 (*PPARγ2*)	F: TTCGGAATCAGCTCTGTGGA
		R: CCATTGGGTCAGCTCTTGTG
	Peroxisome proliferator-activated receptor gamma coactivator 1-alpha (*PGC1α*)	F: TAAATCTGCGGGATGATGGA
		R: GTTTCGTTCGACCTGCGTAA
	PR domain containing 16 (*PRDM16*)	F: GGACCTTTTTGACAGCAGCA
		R: GGGGGCAAAGCATTTAACTC
	Cell death activator CIDE-A (*Cidea*)	F: GGAATCTGCTGAGGTTTATG
		R: ATCCCACAGCCTATAACAGA
	Uncoupling protein 1 (*UCP1*)	F: GGTTTGCACCACACTCCTG
		R: ACATGGACATCGCACAGCTT
	Glyceraldehyde-3-phosphate dehydrogenase (*GAPDH*)	F: GTGATGGCATGGACTGTGGT
		R: GGAGCCAAAAGGGTCATCAT

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
