# Peer review of "Filbertone Ameliorates Adiposity in Mice Fed a High-Fat Diet via Activation of cAMP Signaling"

_nutrients, 2019, doi:10.3390/nu11081749_

Round 1

Reviewer 1 Report

In this study, Youna Moon and colleagues investigated the preventive effects of fibertone on lipid accumulation in the adipose tissue of mice fed a high-fat diet and explored the underlying mechanisms. They found that filbertone supplementation ameliorated adiposity in mice fed a high-fat diet, and such effect might be associated with the activation of cAMP signaling. Overall, the work is of interesting and the manuscript is written well. However, some points should be addressed.

1)      In Abstract, “Taken together, our results suggest that the beneficial effects of filbertone on lipid accumulation may be associated with increased cAMP signaling.” should be revised as “Taken together, our results suggest that the beneficial effects of filbertone on lipid accumulation may be associated with the activation of cAMP signaling.”

2)      The conclusive results in Figure 5 should be shown in Abstract.

3)      In Materials and Methods (Line 152), “50 mL of DMSO” is wrong, and should be “50 μL of DMSO for each well”.

4)      In Figure 5, the authors showed that filbertone reduces triglyceride accumulation in 3T3-L1 cells via cAMP pathway using the adenylate cyclase inhibitor, SQ22,536.  However, the evidence presented in the manuscript was not enough. The authors should also determine the mRNA and protein levels of several key genes (in Figure 3 and Figure 4), which could confirm the in vivo results of filbertone.  

Author Response

Manuscript Number: nutrients-547848

Title: Filbertone ameliorates adiposity in mice fed a high-fat diet via activation of cAMP signaling

Dear Editor-in-Chief:

We sincerely thank the reviewer for his/her positive comments about the importance of and novelty of our work and providing highly constructive suggestions. We have carefully considered all of the comments made by the reviewer, and revised the manuscript accordingly. I have listed each point raised, and then indicated the changes and locations where the revision occurred in the revised manuscript. Corrections were made and highlighted in red color in a revised manuscript.

Reviewer #1 (Comments to the Author):

In this study, Youna Moon and colleagues investigated the preventive effects of fibertone on lipid accumulation in the adipose tissue of mice fed a high-fat diet and explored the underlying mechanisms. They found that filbertone supplementation ameliorated adiposity in mice fed a high-fat diet, and such effect might be associated with the activation of cAMP signaling. Overall, the work is of interesting and the manuscript is written well. However, some points should be addressed.

1. In Abstract, “Taken together, our results suggest that the beneficial effects of filbertone on lipid accumulation may be associated with increased cAMP signaling.” should be revised as “Taken together, our results suggest that the beneficial effects of filbertone on lipid accumulation may be associated with the activation of cAMP signaling.”

Answer: Appreciating the reviewer for this constructive comment, authors corrected the word “ increased cAMP signaling” to the activation of cAMP signaling.” (Line 26)

2. The conclusive results in Figure 5 should be shown in Abstract.

Answer: As suggested by the reviewer, authors newly included the conclusive results in Figure 5 in the abstract section as follows: “Filbertone reduced intracellular lipid accumulation and increased increased oxygen consumption rate in 3T3-L1 cell and these filbertone-induced changes were abrogated by the Adcy inhibitor.” (Line 22-24)

3. In Materials and Methods (Line 152), “50 mL of DMSO” is wrong, and should be “50 μL of DMSO for each well”.

Answer: As commented by the reviewer, “50 mL of DMSO” was corrected to “50 μL of DMSO”. (Line 155)

4. In Figure 5, the authors showed that filbertone reduces triglyceride accumulation in 3T3-L1 cells via cAMP pathway using the adenylate cyclase inhibitor, SQ22,536. However, the evidence presented in the manuscript was not enough. The authors should also determine the mRNA and protein levels of several key genes (in Figure 3 and Figure 4), which could confirm the in vivo results of filbertone.

Answer: We thank the reviewer for raising this insightful comments. Roles of cAMP signaling in the regulation of adipogenesis or thermogenesis have been reported repeatedly in both adipose tissues and adipocytes (Cao W, et al., 2004; Kim N-J, et al., 2019; Li F, et al., 2008; Meng W, et al., 2017), although the exact down-stream signaling pathways need further consensus. In this article, authors revealed that filbertone administration led to the reprogramming of cAMP-mediated signaling pathways involved in adipogenesis and thermogenesis in the adipose tissue of mice fed HFD. The signal map shown in Figure 6 is the proposed mechanism behind the protective effects of filbertone against adiposity in mice based on our in vivo results. Then, the authors decided to conduct the in vitro study to prove whether these effects of filbertone ameliorating adiposity occur through the regulation of cAMP, which cannot be attained from the in vivo study. We show that filbertone reduces triglyceride accumulation and increases oxygen consumption rates in 3T3-L1 cells via cAMP pathway using the adenylate cyclase inhibitor, SQ22,536. Authors would like to emphasize that the main reason we used 3T3-L1 cells in the present study was to uncover the direct involvement of cAMP in filbertone-induced metabolic regulation, rather than identifying the exact cAMP-mediated signaling pathways affected by filbertone in 3T3-L1 cells or confirming the in vivo results (mRNA and protein levels of several key genes in Figure 3 and Figure 4) in in vitro system. Authors would like to ask your favor considering that this is the first report revealing the biological function of filbertone in obesity and could lead to future studies identifying the molecular target (and downstream signaling molecules) of this compound in adipocytes.

- Cao W, et al., p38 Mitogen-activated protein kinase is the central regulator of cyclic AMP-dependent transcription of the brown fat uncoupling protein 1 gene. Molecular and Cellular Biology 24 (2004): 3057-3067.

- Kim N-J, et al., A PDE1 inhibitor reduces adipogenesis in mice via regulation of lipolysis and adipogenic cell signaling. Experimental & Molecular Medicine 51 (2019): 5.

- Li F, et al., Protein kinase A suppresses the differentiation of 3T3-L1 preadipocytes. Cell Research 18 (2008): 311-323.

- Meng W, et al., Rheb inhibits beiging of white adipose tissue via PDE4D5-dependent downregulation of the cAMP-PKA signaling pathway. Diabetes 66 (2017): 1198-1213.

The authors sincerely thank the reviewer for carefully reading our manuscript and providing critical comments. We believe that the reviewer’s comments have resulted in a significantly improved version of our manuscript.

Sincerely yours,

Taesun Park, Ph.D

Department of Food and Nutrition, Yonsei University,

50 Yonsei-ro, Seodaemun-gu, Seoul 120-749, Korea

Phone: +82-2-2123-3123

Fax: +82-2-365-3118

E-mail: tspark@yonsei.ac.kr

Reviewer 2 Report

Manuscript entitled “Filbertone ameliorates adiposity in mice fed a high-fat diet via activation of cAMP signaling” presents important and valuable results of the studies performed on C57BL/6N mice, as well as differentiated mice 3T3L1 adipocytes. It is already known that diet related diseases, especially obesity, result of  improper diet and lack of exercise. According to the WHO about 13% of the world adult population are obese and 39% are overweight, so these metabolic diseases are referred as epidemics of XXI century. To help manage these disturbances there are recommended differently acting substances able to modulate lipids and carbohydrates metabolism; however, due to the variety of complications, less harmful plant derived substitutes are being searched intensively. In presented manuscript authors studied biological activity of filbertone, a naturally occurring flavoring agent, as obesity preventive phytocompound. Obtained results are very promising, however, there are some suggestions that may improve the quality of the paper presented, thus the manuscript needs major revision and more detailed explanation.

1. There is an explanation needed for in vivo chosen filbertone dosage described as 0.025%. Authors shortly mentioned that previously treated mice with 10, 25, 30 and 50 mg/(kg·d) of filbertone for 28  days. However, presented studies were performed for 14 weeks. Please explain why the 0.025% dosage of supplementation was chosen. Were experimental results obtained for treatment for 28 days and 14 weeks comparable?

2. In the methodology section there should be described for how long the experiment with the mice was performed. Additionally the cAMP assay is described with details thus it is suggested to describe other biochemical analysis in a comparable way. Statistics description should be completed.  

3. In the text there are many short names, like PPARγ. It is suggested to explain full names for their first introduction.

4. In the description of Figure 6 it is suggested to explain molecular mechanism of proposed filbertone activity, especially in terms of presented proteins involvement.

5. What was the detailed method for lipid droplets staining?

6. Did authors perform studies on mice with normal diet and filbertone? It shoul be explained directly due to the legends in Figures 1-4, which contain filbertone instead of HFD+filbertone.

7. More detailed information about in vitro studies methodology is needed. How long was the 3T3L1 cells differentiation process peformed (in the manuscript it is mentioned that for 2 days)? What are formulations of differentiation and adipocytes maintenance media (in the manuscript they are the same)? What is the incubation time of 3T3L1 cells with filbertone for MTT assay (it is described as 24 hours)? What is the time of incubation of cells with filbertone before other analysis performance, like mitochondrial oxygen consumption rate? Further in the taxt it was written that cells differentiation was performed for 8 days.

8. In Figure 5abc there are unknown units of filbertone studied dosages. Explain why the dosage 50 µM filbertone instead of 100 µM was chosen? The last one did not reveal toxicity against cells as it is presented in Fig. 5 a. In Figure 5 d (line 245) the authors mentioned transfected cells - what did they mean exactly?

9. The authors studied molecular mechanism of filbertone on cells isolated from mice. Did they observe comparable activity in 3T3L1 cells?

10. In Figures 1-4 authors present data for control mice as ”chow  - - -„. What was the diet for control animals?

11. In lines 172-173 the authors wrote that „Filbertone-treated mice were lighter than those fed the HFD alone (Figure 1A, B) due to reduced 174 visceral fat-pad weights (Figure 1D) that occurred without a change in food intake (Figure 1C)." However in the Figure 1c the intake of food in g does not corelate with that information, it is lower.

12. The authors wrote (289-294): „To identify the molecular targets of filbertone, we researched the molecular diversions occurring in adipose  tissue in response to filbertone treatment using molecular neuroimaging (MNI), a validated method  for the recognition of targets and related signaling of molecules. The results led to the recognition of various ORs including (Olfr 1034, 402, 508, 110) as promising candidates. Identifying the specific OR accountable for the filbertone-induced elevate in cAMP levels in adipose tissue is an significant and  challenging area for future work.” It should be explained with more detailes.

13. English should be checked again.

In summary, manuscript presents valuable information, however it requires major revision.

Author Response

Manuscript Number: nutrients-547848

Title: Filbertone ameliorates adiposity in mice fed a high-fat diet via activation of cAMP signaling

Dear Editor-in-Chief:

We sincerely thank the reviewer for his/her positive comments about the importance of and novelty of our work and providing highly constructive suggestions. We have carefully considered all of the comments made by the reviewer, and revised the manuscript accordingly. I have listed each point raised, and then indicated the changes and locations where the revision occurred in the revised manuscript. Corrections were made and highlighted in red color in a revised manuscript.

Reviewer #2 (Comments to the Author):

Manuscript entitled “Filbertone ameliorates adiposity in mice fed a high-fat diet via activation of cAMP signaling” presents important and valuable results of the studies performed on C57BL/6N mice, as well as differentiated mice 3T3L1 adipocytes. It is already known that diet related diseases, especially obesity, result of improper diet and lack of exercise. According to the WHO about 13% of the world adult population are obese and 39% are overweight, so these metabolic diseases are referred as epidemics of XXI century. To help manage these disturbances there are recommended differently acting substances able to modulate lipids and carbohydrates metabolism; however, due to the variety of complications, less harmful plant derived substitutes are being searched intensively. In presented manuscript authors studied biological activity of filbertone, a naturally occurring flavoring agent, as obesity preventive phytocompound. Obtained results are very promising, however, there are some suggestions that may improve the quality of the paper presented, thus the manuscript needs major revision and more detailed explanation.

1. There is an explanation needed for in vivo chosen filbertone dosage described as 0.025%. Authors shortly mentioned that previously treated mice with 10, 25, 30 and 50 mg/(kg·d) of filbertone for 28 days. However, presented studies were performed for 14 weeks. Please explain why the 0.025% dosage of supplementation was chosen. Were experimental results obtained for treatment for 28 days and 14 weeks comparable?

Answer: We thank the reviewer for raising this insightful comment. Authors added the following sentences to explain why we chose 0.025% dosage of supplementation: “Oral administration dose of 25 mg filbertone/ kg body weight equals 1 mg filbertone/ mouse(40 g body weight). Assumming that mice consumed 1 mg filbertone in 4 g diet, the diet should contains 0.025% filbertone.” (Line 294-296). In the present study, mice fed the 0.025% filbertone-supplemented HFD diet for 14 weeks showed significant reductions in final body weight as compared with HFD‐fed mice. These results are comparable to the result from mice orally administered 25 mg filbertone/kg body weight for 4 weeks.

2. In the methodology section there should be described for how long the experiment with the mice was performed. Additionally the cAMP assay is described with details thus it is suggested to describe other biochemical analysis in a comparable way. Statistics description should be completed.

Answer: We thank the reviewer for pointing these out. Per reviewer’s suggestion, authors corrected each issue in the method section as follows:

-       Authors specified the experimental feeding period of animal study in the method section as follows: “The mice were subjected to the experimental diet for 14 weeks, during which time all animals were permitted ad libitum access to the diet and water.”(Line 82-83)

-       Biochemical analysis details were included in the last sentence under 2.5 Biochemical analysis: Plasma was collected by centrifugation (15 min, 4000 g, 4°C) in EDTA-coated tube. All plasma samples were snap frozen and stored at −80°C.(Line 106-107) The plasma levels of glucose, free fatty acids, triglycerides and total cholesterol were determined using commercially available kits (Bio-Clinical System, Gyeonggi-do, Korea). The plasma concentrations of insulin, tumor necrosis factor-alpha (TNF-α), monocyte chemoattractant protein 1 (MCP1), interleukin 6 (IL-6), adiponectin, and leptin were estimated using a commercially available Enzyme-linked immunosorbent assay (ELISA) kit (Millipore, Burlington, MA, USA). The absorbance was measured using a microplate spectrophotometer (Tecan, Männedorf, Switzerland). (Line 112-113)

-       For more complete description of statistical analyses, following sentences were added: “Data are presented as means ± SEM. Data describing body weight gains, visceral fat‐pad weights, plasma characteristics, and OGTT are presented as means ± SEM of eight mice per group. The RT‐PCR and Western blot results are means from an n = 8 ± SEM of three independent experiments (n = 2 or 3 per experiment) for each group. A p-value 0.05 was considered statistically significant and significance was set at * p < 0.05, ** p < 0.01 and *** p < 0.001. (Line 184-189)

3. In the text there are many short names, like PPARγ. It is suggested to explain full names for their first introduction.

Answer: As guided by the reviewer, we have given the full name of C/EBPα, aP2, Pparγ2, Prdm16, Cidea and UCP1 for their first introduction (Line 233-235, 238-239).

4. In the description of Figure 6 it is suggested to explain molecular mechanism of proposed filbertone activity, especially in terms of presented proteins involvement.

Answer: We thank the reviewer for raising this insightful comment. Authors included the explanation for molecular mechanism of proposed filbertone activity in Figure 6 as follows: The proposed mechanism behind the protective effects of filbertone against adiposity in mice. High cytosolic levels of cAMP induced by filbertone activate PKA and thus cause phosphorylation of AMPK and CREB. Filbertone‐induced AMPK activation inhibits adipocyte differentiation and blocks the expression of late adipogenic markers such as FAS and transcription factors, PPAR‐γ2 and CEBPα, thereby inhibiting adipogenesis. Filbertone‐induced CREB activation promotes the transcription of genes involved in thermogenesis, e.g., PGC‐1α and UCP1, resulting in increased thermogenesis in adipocytes. (Line 281-287)

5. What was the detailed method for lipid droplets staining?

Answer: We thank the reviewer for this comment. Authors included the details of lipid droplets staining method as follows: “Differentiated 3T3-L1 cells were washed with PBS, fixed with 4% paraformaldehyde solution (Cellnest, Minato, Japan) for 20 mins, and allowed to dry. Lipid droplets in mature adipocytes were then stained with Oil Red O staining solution (0.5% Oil red O in isopropanol) for 20 mins and washed with distilled water. Microscope images were obtained to assess the stained oil droplets. To quantify the staining intensity, isopropanol was added into each well to dissolve the staining and dissolved solution was transferred to another 96‐well plate.” (Line 169-174)   

6. Did authors perform studies on mice with normal diet and filbertone? It should be explained directly due to the legends in Figures 1-4, which contain filbertone instead of HFD+filbertone.

Answer: We thank the reviewer for pointing this out. We realized that the legends of graphs and pictures in Figure 1-4 are misleading indeed since we did not perform studies on mice fed normal diet supplemented with filbertone. The word, “Filbertone” in the legend of graphs and pictures has been now corrected to “HFD+filbertone” (Figure 1-4).

7. More detailed information about in vitro studies methodology is needed. How long was the 3T3L1 cells differentiation process performed (in the manuscript it is mentioned that for 2 days)? What are formulations of differentiation and adipocytes maintenance media (in the manuscript they are the same)? What is the incubation time of 3T3L1 cells with filbertone for MTT assay (it is described as 24 hours)? What is the time of incubation of cells with filbertone before other analysis performance, like mitochondrial oxygen consumption rate? Further in the taxt it was written that cells differentiation was performed for 8 days

Answer: Thank you very much for pointing this out. We now have revised the method section for in vitro experiments, providing more detailed information as follows: “For the differentiation assay, preadipocytes were seeded in 6‐well plates and cultured until 2 days post-confluence. After the 3T3‐L1 cells became confluent, the medium was replaced with DMEM consisting of 10% of fetal calf serum (FBS; Gibco‐BRL), 1% of antibiotic-antimycotic solution, 10 μg/mL bovine insulin, 1 μM dexamethasone (Sigma‐Aldrich, St. Louis, MO, USA), and 0.5 mM isobutylmethylxanthine (Sigma‐Aldrich) with or without filbertone (Sigma‐Aldrich) and SQ22,536 (an Adcy inhibitor, Sigma‐Aldrich). The cells were then maintained in DMEM containing 10% of FBS with 10 μg/mL insulin for another 2 days, followed by culturing with DMEM containing 10% of FBS for additional 4 days with or without filbertone and SQ22,536.” (Line 151-158); To evaluate the toxicity of filbertone, 3T3-L1 preadipocytes were plated into 96-well microtiter plates at a density of 1 × 104 cells/well. After 24 h, the culture medium was replaced by 200 µL serial dilutions (0-150 µM) of filbertone and the cells were incubated for another 24 h. Culture solutions were then removed and replaced by 90 µL of culture medium and 10 µL of filtered MTT solution (10 mg/mL) in phosphate-buffered saline (PBS) to reach a final concentration of 1 mg MTT/mL. After 3 h, the unresponded dye was removed, and then the insoluble formazan crystals were dissolved in DMSO (50 µL/well) and measured spectrophotometrically in a microplate reader (Molecular Devices) at 570 nm.”  (Line 160-167)    

8. In Figure 5abc there are unknown units of filbertone studied dosages. Explain why the dosage 50 µM filbertone instead of 100 µM was chosen? The last one did not reveal toxicity against cells as it is presented in Fig. 5 a. In Figure 5 d (line 245) the authors mentioned transfected cells - what did they mean exactly?

Answer: Thank you very much for your insightful comments. Authors chose 50 µM filbertone concentration based on the results that toxicity was observed from 150 µM, while activity was observed from 25 µM concentrations. We chose the dosage with enough safety margin and assured effectiveness. The word ‘transfected’ is a typographical error and now is corrected to ‘treated’ in the legend of Figure 5d. (Line 273)

9. The authors studied molecular mechanism of filbertone on cells isolated from mice. Did they observe comparable activity in 3T3L1 cells?

Answer: Authors would like to make it clear that all of the in vitro experiments, including molecular mechanism of filbertone, in this article were done in 3T3-L1 cells, not in cells isolated from mice.

10. In Figures 1-4 authors present data for control mice as”chow - - -„. What was the diet for control animals?

Answer: We thank the reviewer for this comment. Throughout the manuscript, the words ‘control mice’ is now corrected to ‘control mice fed the HFD alone’. (Line 194, 212, 241)

11. In lines 172-173 the authors wrote that „Filbertone-treated mice were lighter than those fed the HFD alone (Figure 1A, B) due to reduced 174 visceral fat-pad weights (Figure 1D) that occurred without a change in food intake (Figure 1C)." However in the Figure 1c the intake of food in g does not corelate with that information, it is lower.

Answer: We thank the reviewer for this valuable comment. In order to improve the clarity, the results has been revised as follow: “Mice fed the filbertone for 14 weeks showed significant reductions in final body weight and body weight gain as compared to control mice fed HFD alone, without changes in food intake (Figure 1A-C). The total visceral (epididymal, perirenal, mesenteric, and retroperitoneal) fat pad weights were significantly lower in the filbertone‐fed mice than in the HFD control mice (Figure 1D).”(Line 193-196)

12. The authors wrote (289-294): “To identify the molecular targets of filbertone, we researched the molecular diversions occurring in adipose tissue in response to filbertone treatment using molecular neuroimaging (MNI), a validated method for the recognition of targets and related signaling of molecules. The results led to the recognition of various ORs including (Olfr 1034, 402, 508, 110) as promising candidates. Identifying the specific OR accountable for the filbertone-induced elevate in cAMP levels in adipose tissue is an significant and challenging area for future work.” It should be explained with more detailes.

Answer: Thank you very much for your insightful comments. We noticed that there is an error in describing the full name of MNI, and it is now corrected it to ‘a mode-of-action by network identification (MNI)’ and included the relevant reference for MNI method (Line 328). For detailed explanation, authors newly included the sentences in the discussion as follows: “To identify the molecular targets of filbertone, we researched the molecular diversions occurring in adipose tissue in response to filbertone treatment using a mode-of-action by network identification (MNI), a validated method for the recognition of targets and related signaling of molecules [30,31]. The analysis of gene profiles with highest MNI rankings and fold changes in differential expression led to the recognition of various ORs including Olfr 1034, 402, 508, 110 as promising candidates.” (Line 328-331)

References:

30. Jain, N.; Thatte, J.; Braciale, T.; Ley, K.; O'Connell, M.; Lee, J.K. Local-pooled-error test for identifying differentially expressed genes with a small number of replicated microarrays. Bioinformatics 2003, 19, 1945-1951.

31. Storey, J.D.; Tibshirani, R. Statistical methods for identifying differentially expressed genes in DNA microarrays. In Functional Genomics, Springer: 2003; pp. 149-157.

13. English should be checked again.

Answer: We have rechecked English editing and typographical errors throughout the manuscript.

The authors sincerely thank the reviewer for carefully reading our manuscript and providing critical comments. We believe that the reviewer’s comments have resulted in a significantly improved version of our manuscript.

Sincerely yours,

Taesun Park, Ph.D

Department of Food and Nutrition, Yonsei University,

50 Yonsei-ro, Seodaemun-gu, Seoul 120-749, Korea

Phone: +82-2-2123-3123

Fax: +82-2-365-3118

E-mail: tspark@yonsei.ac.kr

Round 2

Reviewer 2 Report

I am satisfied with improvements and explanations provided by the Authors.